# Use of Next-Generation Sequencing to Support the Diagnosis of Familial Interstitial Pneumonia

**DOI:** 10.3390/genes14020326

**Published:** 2023-01-27

**Authors:** Ana Rita Gigante, Eduarda Milheiro Tinoco, Ana Fonseca, Inês Marques, Agostinho Sanches, Natália Salgueiro, Carla Nogueira, Sérgio Campainha, Sofia Neves

**Affiliations:** 1Serviço de Pneumologia, Centro Hospitalar de Vila Nova de Gaia/Espinho, 4434-502 Vila Nova de Gaia, Portugal; 2Serviço de Imagiologia, Centro Hospitalar de Vila Nova de Gaia/Espinho, 4434-502 Vila Nova de Gaia, Portugal; 3Serviço de Anatomia Patológica, Centro Hospitalar de Vila Nova de Gaia/Espinho, 4434-502 Vila Nova de Gaia, Portugal; 4Departamento de Genética Molecular, Synlab-Genética Médica, 4100-321 Porto, Portugal

**Keywords:** interstitial lung disease, genetics, phenotype

## Abstract

Familial interstitial pneumonia (FIP) is defined as idiopathic interstitial lung disease (ILD) in two or more relatives. Genetic studies on familial ILD discovered variants in several genes or associations with genetic polymorphisms. The aim of this study was to describe the clinical features of patients with suspected FIP and to analyze the genetic variants detected through next-generation sequencing (NGS) genetic testing. A retrospective analysis was conducted in patients followed in an ILD outpatient clinic who had ILD and a family history of ILD in at least one first- or second-degree relative and who underwent NGS between 2017 and 2021. Only patients with at least one genetic variant were included. Genetic testing was performed on 20 patients; of these, 13 patients had a variant in at least one gene with a known association with familial ILD. Variants in genes implicated in telomere and surfactant homeostasis and *MUC5B* variants were detected. Most variants were classified with uncertain clinical significance. Probable usual interstitial pneumonia radiological and histological patterns were the most frequently identified. The most prevalent phenotype was idiopathic pulmonary fibrosis. Pulmonologists should be aware of familial forms of ILD and genetic diagnosis.

## 1. Introduction

Familial interstitial pneumonia (FIP) (MIM: 178500) is usually defined as idiopathic interstitial lung disease (ILD) in two or more relatives who share a common ancestry [1,2,3]. Adults with FIP are practically indistinguishable from sporadic ILD patients concerning their clinical presentation, radiographic findings, and histopathology, except that those with FIP tend to present at an earlier age and with a more aggressive natural course of the disease [1,4,5,6].

A genetic predisposition to interstitial lung disease is proposed because of a 10-fold increase in the prevalence of the disease in families of patients with a diagnosis of idiopathic pulmonary fibrosis (IPF) [3,7,8]. Genetic studies on familial forms of ILD led to the discovery of variants in genes implicated in telomere and surfactant homeostasis or associated with several genetic polymorphisms [1,3,9,10,11]. For most of these genes, many analyses of FIP families have suggested an autosomal dominant pattern of inheritance with incomplete penetrance [4,12,13].

Considering gene variants, telomerase complex mutations result in overall decreased telomerase activity and telomere length, which manifests as premature aging disorders such as pulmonary fibrosis, premature hair graying, bone marrow failure, and/or liver cirrhosis, called short telomere syndrome [1,2,9,14,15,16]. Furthermore, a family history of neonatal respiratory distress or childhood interstitial lung disease is relevant for suggesting the possibility of a surfactant-related disorder [1,2,9]. In addition to monogenic Mendelian diseases, a common polymorphism in the promoter region of the *MUC5* gene increases the risk of idiopathic pulmonary fibrosis, either sporadic or familial [1,9,11].

Genetic testing enables several genes to be analyzed together. For patients with suspected FIP, next-generation sequencing (NGS) allows the analysis of a targeted gene panel with a known association with familial ILD [9]. The current study aimed to describe the clinical features of patients with suspected FIP and to analyze the genetic variants detected through NGS.

## 2. Methods

A retrospective observational analysis was conducted in a patient population followed in the ILD outpatient clinic of Centro Hospitalar de Vila Nova de Gaia/Espinho who had ILD and a family history of ILD in at least one first- or second-degree relative and who underwent NGS genetic testing from a peripheral blood sample between 2017 and 2021. Peripheral blood samples along with consent and family history were taken from the patients. Genomic DNA (gDNA) was extracted from peripheral blood samples using the Chemagic™ 360 automated extractor (PerkinElmer) according to the manufacturer’s instructions. DNA quantity was evaluated using the Quantus™ Fluorometer (Promega, Madison, Wisconsin, USA). DNA was diluted according to the input recommendation of TWIST BioScience library preparation. NGS analysis included a gene panel that allowed the identification of variants in the following genes: ATP-binding cassette member A3, *ABCA3* (NM_001089.2); colony-stimulating factor 2 receptor subunit alpha, *CSF2RA* (NM_006140); colony-stimulating factor 2 receptor subunit beta, *CSF2RB* (NM_000395); dyskerin, *DKC1* (NM_001363.4); mucin 5B, *MUC5B* (NM_002458.2); NK2 Homeobox 1, *NKX2-1* (NM_001079668.2); polyadenylate-specific ribonuclease, *PARN* (NM_002582.3); regulator of telomere elongation helicase, *RTEL1* (NM_032957.4); surfactant protein A1, *SFTPA1* (NM_005411.5); surfactant protein A2, *SFTPA2* (NM_001098668.2); surfactant protein B, *SFTPB* (NM_198843.2); surfactant protein C, *SFTPC* (NM_003018.3); surfactant protein D, *SFTPD* (NM_003019.4); telomerase RNA component, *TERC* (NM_001566.1); telomerase reverse transcriptase, *TERT* (NM_198253.2); and telomere interacting factor 2, *TINF2* (NM_001099274.1). The libraries were prepared with the Twist Human Core Exome Plus kit, and NGS was carried out on an Illumina platform, with more than 99.00% of the bases having coverage greater than 20x. The sequences were aligned with the reference genome (GRCh37/hg19). Bioinformatics analysis was performed with an in-house pipeline using different bioinformatics tools. The analysis of CNVs (copy number variations) was executed using an in-house bioinformatics pipeline. All pathogenic variants, probably pathogenic variants, and variants of uncertain clinical significance were reported. Sanger sequencing was used for the confirmation of the presence of pathogenic or likely pathogenic genetic variants. The variants’ classification followed the current guidelines from the American College of Medical Genetics and Genomics [17].

Only patients with at least one genetic variant identified by NGS were included. Demographic data, the number and kinship of affected family members, clinical, functional, radiological, and histopathological data, genetic testing results, and multidisciplinary diagnosis (considered the phenotype) were retrospectively reviewed from clinical files.

The study protocol was approved by the ethics committee of the Centro Hospitalar de Vila Nova de Gaia/Espinho, and patients or relatives provided informed consent.

Descriptive statistical analyses were carried out using the Statistical Package for the Social Sciences (SPSS)^®^ program, Chicago, Illinois, USA, version 22.0. Categorical variables are described as frequencies (n) and percentages (%), and continuous variables are described using mean ± standard deviation or median and interquartile range, according to the distribution of the data.

## 3. Results

Genetic testing was performed on a total of 20 patients with ILD. Of these, 13 patients (65%) had a variant in at least one gene with a known association with familial ILD, whose characteristics are presented in Table 1. Patients were mostly male (69.2%), with a mean age at symptom onset of 60.5 ± 11.1 years and a mean age at diagnosis of lung disease of 64.4 ± 10.3 years. Seven patients had one relative with ILD, and six patients had two or more; 84.6% of cases were first-degree relatives, and sibling was the predominant kinship (69.2%). A history of smoking was reported in 38.5%, and 69.2% reported exposure to some fibrogenic agents.

The most frequent symptoms reported at presentation were dyspnea (76.9%) and cough (69.2%). Fine crackles on pulmonary auscultation were found in 76.9% of patients. Five patients had at least one extrapulmonary manifestation, in which variants of genes involved in telomere homeostasis were detected. At diagnosis, a probable usual interstitial pneumonia radiological pattern was the most frequently identified (*n* = 5). Histology was obtained from nine patients; a probable usual interstitial pneumonia pattern (*n* = 3) was most frequently observed.

Table 2 shows the genetic variants detected in the 13 patients with suspected FIP. Variants in genes implicated in telomere homeostasis such as *TERT*, *RTEL1*, *PARN*, and *TINF2* were detected in eight (61.5%) patients; four (30.8%) patients presented variants in genes implicated in surfactant homeostasis: *SFTPA2*, *ABCA3*, and *DMBT1*; and two (15.4%) patients had a *MUC5B* variant. Most variants were classified with uncertain clinical significance, and four variants were classified as likely pathogenic. Patient #9 presented variants in two genes: one variant in the *TERT* gene in heterozygosity and two variants in the *ABCA3* gene. The most prevalent phenotype was IPF in 61.5% of patients. Two siblings (patients #5 and #8) presented the same variant in the *RTEL1* gene but with different phenotypes: IPF and fibrotic hypersensitivity pneumonitis, respectively.

Table 3 presents the clinical characteristics, number, and type of relatives affected by the patient’s genetic variants.

## 4. Discussion

Herein, we present the first study, to the best of our knowledge, on familial ILD and genetic testing data in Portugal.

There are no clear guidelines concerning genetic testing and counseling in patients with FIP. Experts recommend a thorough family history in all subjects with idiopathic ILD, regardless of patient age. Genetic testing should be offered to all patients with suspected FIP and recommended in circumstances when a genetic diagnosis is likely to be achieved [1,2,3,9,10]. 

One study revealed that in ILD families, the presence of ILD is more frequent in older male smokers (mean age of 68 years) [4]. We observed a male preponderance but a symptom onset at a younger mean age, and more patients were non-smokers; however, more than half reported some environmental exposure. Indeed, it is suggested that environmental exposure may contribute to developing ILD in individuals genetically prone to this disease [4]. As reported in the literature [1,4,6,10], we detected clinical-radiological heterogeneity within one family, where the same variant resulted in different phenotypes. These data support the importance of the interplay of the genetic environment in the manifestation of this disease. 

We observed an average delay of 4 years from symptom onset to ILD diagnosis. Given the retrospective nature of our study, there are limitations to understanding this delay. The reported delay seems higher than that observed in previous studies. In the INTENSITY survey, the median time from symptom onset to current diagnosis was 7 months (range, 0–252 months) [18]. Pritchard et al. report that in ILD patients, the median time from cough documentation to pulmonology referral is 13.2 months (IQR 1.8–58.9 months) [19]. Another study with a large sample of IPF Medicare beneficiaries revealed that the first chest CT scan was performed in 57.5% of IPF patients more than one year before diagnosis [20]. In our study, we hypothesize that the delay observed could be explained by the nonspecific nature of the symptoms, but also by the COVID-19 pandemic, which hindered access to health care. On other hand, in our outpatient clinics, carrying out genetic tests on a regular basis on patients with a family history of ILD or an IPF diagnosis at a young age is a relatively recent achievement. Apart from that, in some patients, the knowledge of an ILD family history occurred years after their own diagnosis, so genetic studies were only carried out later. In our study, IPF was the predominant phenotype. Studies suggested that familial forms of the disease accounted for 2–20% of IPF cases [1,3,10]. Additionally, the usual interstitial pneumonia radiological and histopathological pattern was the most frequently described in studies of familial ILD [1,4,9,10,21].

Variants in genes implicated in telomere homeostasis were more frequently found: *RTEL1* variants in three patients, *TERT* variants in two patients, *PARN* variants in two patients, and a *TINF2* variant in one patient. Indeed, heterozygous variants have been detected in familial forms of pulmonary fibrosis involving *TERT* (∼15%), *RTEL1* (5–10%), *PARN* (∼5%), and *TERC* (∼3%). Variants in *TINF2* are much rarer [6,14,22,23,24,25]. Of the patients in our study, five presented hematological and/or liver abnormalities and/or premature graying of hair. This observation validates the recommendation of genetic testing in patients with ILD and these extrapulmonary manifestations, as they are indicative of short telomere syndrome [2,3,15,16]. 

For families or patients with a history suggestive of short telomere syndrome, some teams propose the peripheral blood mononuclear cell (PBMC) telomere length test [1,2,9]. If the PBMC telomere length is short (below the 10^th^ percentile for age), the likelihood of identifying a pathogenic variant in a known telomerase-related gene is high. Of note, the transmission of the telomere length is independent of the transmission of the mutation, and thus, it is possible to inherit short telomeres (and disease risk) without inheriting a mutation; therefore, this assessment can complement the genetic study [2,9,15,26,27]. Unfortunately, in our center, telomere length measurement is not available. This assessment gains particular importance in the case of young patients proposed for lung transplantation. Indeed, in an independent cohort, patients with telomere lengths below the 10th percentile before transplant were reported to have worse survival and a shorter time to the onset of chronic lung allograft dysfunction [28].

The variants’ classification followed the current guidelines from the American College of Medical Genetics and Genomics, which provide a framework for reporting and interpreting the pathogenicity of genetic variants: pathogenic, likely pathogenic, variant of uncertain significance, likely benign, and benign [17]. This classification is based on criteria using typical types of variant evidence when available, such as the nature of the variant, previous reports, population data, segregation, functional and computational data, and the specific phenotype [9,17]. Most variants were classified as variants of uncertain significance, although this classification may change with emerging evidence [17]. Four probably pathogenic heterozygous variants were detected in genes with autosomal dominant transmission and, therefore, were most likely responsible for the disease [3]. One patient had two variants detected in possible compound heterozygosity in a gene with autosomal recessive transmission (*ABCA3* gene), which could contribute to the disease [3]. However, as we did not perform a parental study, it is impossible to conclude whether the variants were on different alleles [17]. 

In seven patients with suspected FIP, no genetic variants were detected. Of note, there are limitations to genetic testing using NGS. Several types of genetic variants are not robustly detected by NGS methods, such as expansions, complex rearrangements, variants in guanine–cytosine-rich regions, or some copy number variants, and there may be novel variants or genes yet to be discovered [29]. Furthermore, gene sequencing only detects variants in known telomerase-related genes without a telomere length assessment, which is particularly significant in suspected short telomere syndrome [2]. Moreover, the genes included in our NGS panel changed over time with the addition of new genes, so there were genes not included in earlier NGS panels from older cases, which is a limitation of the study (see Appendix A for the gene panel analyzed by NGS in each patient).

Despite the retrospective single-center nature and its small sample size, this study provides the first data on the genetic evaluation of a Portuguese population with familial ILD. It might be of interest to create a multicentric database to record the genetic variants of Portuguese FIP cases.

## 5. Conclusions

To finalize, pulmonologists should be conscious of familial forms of ILD and genetic diagnoses. Because a positive genetic diagnosis may carry prognostic value, promote risk stratification in lung transplant candidates, and aid in risk estimation for close relatives, genetic testing may be of benefit in patients with FIP and a high likelihood of a positive genetic diagnosis [2,3]. Lastly, further studies are needed to understand the impact of the genetic background on disease progression to better guide personalized therapeutic options for these patients.

## Figures and Tables

**Table 1 genes-14-00326-t001:** Characteristics of patients with suspected FIP.

Characteristics	N = 13
Basal characteristics at diagnosis	
Sex (male)	9 (69.2)
Age at symptom onset, years	60.5 ± 11.1
Age at diagnosis, years	64.4 ± 10.3
Body mass index, Kg/m^2^	26 [25–29]
Number of family members with ILD 1 2 3 4	7 (53.8) 4 (30.8) 1 (7.7) 1 (7.7)
Degree and kinship of affected relative First degree Sibling(s) >1 sibling Parent Second degree Cousin(s) Aunt/Uncle(s)	9 (69.2) 4 2 (15.4) 3 (23.1) 1 (7.7)
Smoking habits Never smoker Current smoker Former smoker	8 (61.5) 3 (23.1) 2 (15.4)
Fibrogenic exposures Wood Metals Cement Silica Perchloroethylene Cotton Mold Birds	9 (69.2) 2 1 2 3 1 1 2 2
Clinical characteristics at diagnosis	
Signs and symptoms Dyspnea Cough Chest pain Wheeze Expectoration Hemoptysis Weight loss Asthenia Pneumonia Pneumothorax Fine crackles Digital clubbing	10 (76.9) 9 (69.2) 1 (7.7) 2 (15.4) 5 (38.5) 2 (15.4) 1 (7.7) 1 (7.7) 1 (7.7) 0 (0) 10 (76.9) 3 (23.1)
Extrapulmonary manifestations Hematological involvement Macrocytosis and thrombocytopenia Liver involvement Unexplained elevated liver enzymes Idiopathic cirrhosis Mucocutaneous involvement Premature hair graying (before the age of 30)	6 (46.2) 1 (7.7) 1 2 (15.4) 1 1 3 (23.1) 3
Comorbidities Pulmonary hypertension Emphysema OSAS GERD Obesity	0 (0) 2 (15.4) 1 (7.7) 2 (15.4) 3 (23.1)
Functional characteristics at diagnosis	
FVC (% predicted)	75.1 ± 9.8
FEV_1_ (% predicted)	75.1 ± 10.1
FEV_1_/FVC	79.1 ± 8.0
TLC (% predicted)	80.5 ± 12.2
D_LCO_ (% predicted)	54.9 ± 14.4
PaO_2_, mmHg	77.7 ± 11.7
PaCO_2_, mmHg	39.4 ± 4.2
6 min walking test Distance, m Initial SpO_2_ Final SpO_2_ Desaturation (%) Initial HR, bpm Final HR, bpm	465.3 ± 141.4 94.8 ± 1.8 90.5 ± 3.6 4.5 ± 3.2 82.8 ± 17.5 100 [92–118]
Radiological characteristics at diagnosis	
Chest HRCT pattern UIP Probable UIP Indeterminate UIP Fibrotic HP NSIP CFPE	1 (7.7) 5 (38.5) 1 (7.7) 2 (15.4) 2 (15.4) 2 (15.4)
Histopathological characteristics	
Lung biopsy performed Surgical lung biopsy Transbronchial lung cryobiopsy	9 (69.2) 2 7
Histopathological pattern, (N = 9) UIP Probable UIP Fibrotic HP NSIP Unclassifiable	2 (22.2) 3 (33.3) 1 (11.1) 1 (11.1) 2 (22.2)

Abbreviations: CPFE, combined pulmonary fibrosis and emphysema; DLCO, diffusing capacity for carbon monoxide; GERD, Gastroesophageal Reflux Disease; HP, hypersensitivity pneumonitis; HR, Heart Rate; HRCT, high-resolution computed tomography; FEV1, forced expiratory volume in 1 s; FVC, forced vital capacity; NSIP, nonspecific interstitial pneumonia; OSAS, obstructive sleep apnea syndrome; PaCO_2_, partial pressure of carbon dioxide in arterial blood; PaO_2_, partial pressure of oxygen in arterial blood; SpO_2_, peripheral oxygen saturation; TLC, total lung capacity; UIP, usual interstitial pneumonia. Data are presented as n (%), mean ± standard deviation, or median [interquartile range].

**Table 2 genes-14-00326-t002:** Genetic variants detected by NGS in patients with suspected FIP.

Gene	Patient	Variant	Status	Variant Classification	Phenotype ^a^
*TERT*	#2	c.1072C>T	Heterozygosity	Uncertain significance	IPF
	#9	c.2701C>T	Heterozygosity	Uncertain significance	Idiopathic NSIP
*RTEL1*	#5	c.3775_3776del ^b^	Heterozygosity	Likely pathogenic	IPF ^b^
	#8	c.3775_3776del ^b^	Heterozygosity	Likely pathogenic	Fibrotic HP ^b^
	#13	c.2672C>T	Heterozygosity	Uncertain significance	IPF
*PARN*	#11	c.1500T>G	Heterozygosity	Likely pathogenic	CPFE
	#12	c.24dup	Heterozygosity	Likely pathogenic	IPF
*TINF2*	#7	c.1285C>A	Heterozygosity	Uncertain significance	CPFE
*SFTPA2*	#1	c.135C>T	Heterozygosity	Uncertain significance	IPF
*ABCA3*	#4	c.694C>T	Heterozygosity	Uncertain significance	IPF
	#9	c.2026G>A c.1417G>A	Heterozygosity ^c^ Heterozygosity ^c^	Uncertain significanceUncertain significance	Idiopathic NSIP
*DMBT1*	#3	c.3052T>A	Heterozygosity	Uncertain significance	IPF
*MUC5B*	#10	c.1855C>T	Heterozygosity	Uncertain significance	Fibrotic HP
	#6	c.9563C>T	Heterozygosity	Uncertain significance	IPF

Abbreviations: CPFE, combined pulmonary fibrosis and emphysema; IPF, idiopathic pulmonary fibrosis; HP, hypersensitivity pneumonitis; NSIP, nonspecific interstitial pneumonia. ^a^ The multidisciplinary diagnosis was considered the phenotype. ^b^ Two siblings with the same variant but different phenotypes. ^c^ Possible compound heterozygosity, but it is impossible to conclude whether the variants were located on different alleles (in trans); it would be necessary to carry out family segregation studies and re-analyze panels.

**Table 3 genes-14-00326-t003:** Characteristics of patients by genetic variant.

Patient	Genetic Variant	Radiological Pattern	Histological Pattern	Relevant Comorbidities	Number of Relatives with ILD	Degree and Kinship of Affected Relative
#1	*SFTPA2* c.135C>T	Probable UIP	UIP	Obesity	2	1st degree Siblings
#2	*TERT* c.1072C>T	Indeterminate UIP	Probable UIP	GERD Obesity	1	1st degree Sibling
#3	*DMBT1* c.3052T>A	UIP	N/A	-	2	1st degree Siblings
#4	*ABCA3* c.694C>T	NSIP	Unclassifiable	-	2	1st degree Siblings
#5	*RTEL1* c.3775_3776del	NSIP	UIP	Unexplained elevated liver enzymes	1	1st degree Sibling
#6	*MUC5B* c.9563C>T	Probable UIP	N/A	OSAS Obesity	1	1st degree Sibling
#7	*TINF2* c.1285C>A	CPFE	Unclassifiable	Emphysema	1	2nd degree Cousin
#8	*RTEL1* c.3775_3776del	Fibrotic HP	Fibrotic HP	Macrocytosis and thrombocytopenia Idiopathic cirrhosis	1	1st degree Sibling
#9	*TERT* c.2701C>T *ABCA3* c.2026G>A c.1417G>A	Probable UIP	NSIP	Premature hair graying	2	1st degree Siblings
#10	*MUC5B* c.1855C>T	Fibrotic HP	N/A	GERD	1	1st degree Siblings
#11	*PARN* c.1500T>G	CPFE	Probable UIP	Emphysema Premature hair graying	4	1st degree Parent 2nd degree Cousins Aunt
#12	*PARN* c.24dup	Probable UIP	N/A	-	3	2nd degree Cousins
#13	*RTEL1* c.2672C>T	Probable UIP	Probable UIP	Premature hair graying	1	1st degree Parent

Abbreviations: CPFE, combined pulmonary fibrosis and emphysema; GERD, Gastroesophageal Reflux Disease; HP, hypersensitivity pneumonitis; ILD, interstitial lung disease; N/A, not available; NSIP, nonspecific interstitial pneumonia; OSAS, obstructive sleep apnea syndrome; UIP, usual interstitial pneumonia.

## Data Availability

Not applicable.

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
