# Peer review of "Use of Next-Generation Sequencing to Support the Diagnosis of Familial Interstitial Pneumonia"

_genes, 2023, doi:10.3390/genes14020326_

Round 1

Reviewer 1 Report

In this paper Gigante and colleagues described a small but significative population of 20 patients with Familiar Interstitial Pneumonia (FIP). The population was evaluated with a NGS technology to search variants in 16 genes known to be associated to pulmonary fibrosis. The authors also presented demographic and clinical data of the 13 patients in which a genetic variant was found. The majority of patients presented only variants of uncertain significance.

The study is overall well written and explained. I would rise some issues that, if solved, could emphasize the relevance of this paper.

- Firstly, the major limitation of the study as stated by the authors is that not all the patients were tested for all the genes described in the methods section. Authors should add a supplement with a table describing which genes were evaluated for every patient.

- A table describing every patient singularly could be of interest to have a complete look regarding genetic variant, radiological and histological patterns, comorbidities, number and kind of relatives with FIP.

- The data presented in table 1 show a 4 year delay on ILD diagnosis, this problem is well known in literature (see https://doi.org/10.1186/s12931-019-1228-2; https://doi.org/10.1016/j.rmed.2006.10.002; doi: 10.1513/AnnalsATS.201806-376RL.; https://doi.org/10.1186/s12890-017-0560-x)  Authors should discuss it in the paper, trying to explain this delay in patients presenting a familiar form of pathology (so probably already diagnosed relatives)

- Authors should recontrol Table 1 in the extrapulmonary presentations: 1 patient presented am hematological involvement but 2 different presentation are listed; 3 patients presented liver involvement but only 2 presentations are listed.

- Authors should add to the methods section a description of the sample used and the methodology to extract and analyze the DNA, or at least a citation to a paper describing the methodology used, to ensure reproducibility.

Author Response

Thank you for giving us the opportunity to submit a revised draft of our manuscript entitled “Use of Next-Generation Sequencing to support the diagnosis of Familial Interstitial Pneumonia”, to Genes. We appreciate the time and effort that you have dedicated to providing your valuable feedback on our manuscript. We have been able to incorporate changes to reflect most of the suggestions. We have highlighted the changes within the manuscript.

Here is a point-by-point response to the reviewer’s comments and concerns.

Comments from Reviewer: 1

In this paper Gigante and colleagues described a small but significative population of 20 patients with Familiar Interstitial Pneumonia (FIP). The population was evaluated with a NGS technology to search variants in 16 genes known to be associated to pulmonary fibrosis. The authors also presented demographic and clinical data of the 13 patients in which a genetic variant was found. The majority of patients presented only variants of uncertain significance.

The study is overall well written and explained. I would rise some issues that, if solved, could emphasize the relevance of this paper.

- Firstly, the major limitation of the study as stated by the authors is that not all the patients were tested for all the genes described in the methods section. Authors should add a supplement with a table describing which genes were evaluated for every patient.

We would like to thank you for your comments.

We agree and we have clarified this information in the supplementary material.

- A table describing every patient singularly could be of interest to have a complete look regarding genetic variant, radiological and histological patterns, comorbidities, number and kind of relatives with FIP.

We appreciate the advice and we have presented this data in the supplementary material.

- The data presented in table 1 show a 4 year delay on ILD diagnosis, this problem is well known in literature (see https://doi.org/10.1186/s12931-019-1228-2; https://doi.org/10.1016/j.rmed.2006.10.002; doi: 10.1513/AnnalsATS.201806-376RL.; https://doi.org/10.1186/s12890-017-0560-x)  Authors should discuss it in the paper, trying to explain this delay in patients presenting a familiar form of pathology (so probably already diagnosed relatives)

We would like to thank the reviewer for pointing out these interesting works. We included this question in our discussion.

- Authors should recontrol Table 1 in the extrapulmonary presentations: 1 patient presented am hematological involvement but 2 different presentation are listed; 3 patients presented liver involvement but only 2 presentations are listed.

This observation is correct. We have revised it.

- Authors should add to the methods section a description of the sample used and the methodology to extract and analyze the DNA, or at least a citation to a paper describing the methodology used, to ensure reproducibility.

We have added this information to the manuscript.

We would like to thank the reviewer again for taking the time to review our manuscript.

Sincerely,

Ana Rita Gigante

10-Jan-2023

Reviewer 2 Report

Thank you for the opportunity to review this manuscript. The authors describe a retrospective cohort of 20 patients presenting to a single center in Portugal between 2017 and 2021 with apparent familial interstitial lung disease. Of these, 13 had variants detected on a next-generation sequencing (NGS) panel of genes associated with ILD. The most commonly implicated genes were those involved in telomere length, surfactant homeostasis, and MUC5B. Histology was performed on 9 patients and usual interstitial pneumonia was the most commonly-observed pattern. Most patients were “never smokers” and had a sibling with ILD. Average age on presentation was 60.5 years.

There are a number of limitations to this study that reduce its potential scientific impact:

-          The genes included in the panel changed over the course of the study, meaning that patients were not all evaluated with the same instrument.

-          Although 8 patients had variants in genes associated with shortened telomeres, telomere length itself was not assessed.

-          Alleleic segregation within families was not affected for patients with variants of unknown significance.

-          Immunohistochemistry and/or electron microscopy for lamellar body ultrastructure would have informed phenotyping the two patients with variants in ABCA3, and would have allowed for more accurate determination of pathogenicity of the variants.

-          Reflex to whole genome/exome sequencing was not performed for patients with no variants identified with the NGS panel.

Addressing any of these would significantly improve the quality of this report. Despite its limitations, this paper does present data that are from a previously undescribed population. The authors state that this is the first such series from Portugal, and this is consistent with my own search results. This represents the primary interest in this manuscript, as other studies have published more detailed data from larger cohorts.

Minor revisions recommended:

-          There is a typographical error on line 146: “…to this disease4.”

-          Rather than using the term “mutations,” I suggest using “variants” with qualifiers such as “of unknown significance”, “likely pathogenic”, etc.

Author Response

Thank you for giving us the opportunity to submit a revised draft of our manuscript entitled “Use of Next-Generation Sequencing to support the diagnosis of Familial Interstitial Pneumonia”, to Genes. We appreciate the time and effort that you have dedicated to providing your valuable feedback on our manuscript. We have been able to incorporate changes to reflect most of the suggestions. We have highlighted the changes within the manuscript.

Here is a point-by-point response to the reviewer’s comments and concerns.

Comments from Reviewer: 2

Thank you for the opportunity to review this manuscript. The authors describe a retrospective cohort of 20 patients presenting to a single center in Portugal between 2017 and 2021 with apparent familial interstitial lung disease. Of these, 13 had variants detected on a next-generation sequencing (NGS) panel of genes associated with ILD. The most commonly implicated genes were those involved in telomere length, surfactant homeostasis, and MUC5B. Histology was performed on 9 patients and usual interstitial pneumonia was the most commonly-observed pattern. Most patients were “never smokers” and had a sibling with ILD. Average age on presentation was 60.5 years.

There are a number of limitations to this study that reduce its potential scientific impact:

-          The genes included in the panel changed over the course of the study, meaning that patients were not all evaluated with the same instrument.

-          Although 8 patients had variants in genes associated with shortened telomeres, telomere length itself was not assessed.

-          Alleleic segregation within families was not affected for patients with variants of unknown significance.

-          Immunohistochemistry and/or electron microscopy for lamellar body ultrastructure would have informed phenotyping the two patients with variants in ABCA3, and would have allowed for more accurate determination of pathogenicity of the variants.

-          Reflex to whole genome/exome sequencing was not performed for patients with no variants identified with the NGS panel.

Addressing any of these would significantly improve the quality of this report. Despite its limitations, this paper does present data that are from a previously undescribed population. The authors state that this is the first such series from Portugal, and this is consistent with my own search results. This represents the primary interest in this manuscript, as other studies have published more detailed data from larger cohorts.

In fact, it is an observational retrospective study that reflects the clinical practice of a Portuguese hospital. These recommendations are very interesting and pertinent, but we don't have the resources to carry them out. However, it will be considered in future studies.

Minor revisions recommended:

-          There is a typographical error on line 146: “…to this disease4.”

We would like to thank you for your comments.

We have fixed the error.

-          Rather than using the term “mutations,” I suggest using “variants” with qualifiers such as “of unknown significance”, “likely pathogenic”, etc.

We agree and we have changed.

We would like to thank the reviewer again for taking the time to review our manuscript.

Sincerely,

Ana Rita Gigante

10-Jan-2023

Round 2

Reviewer 1 Report

I appreciate the work of authors on the revised paper.

Some minor details:

-In table S2 CPFE is misspelled

-I would like to see table S2 in the paper and not in the supplement if the authors have space in the paper

Author Response

We would like to thank you for your comments.

-In table S2 CPFE is misspelled

We have fixed the error.

-I would like to see table S2 in the paper and not in the supplement if the authors have space in the paper

We have added this table to the manuscript as Table 3.

We would like to thank the reviewer again for taking the time to review our manuscript.

Sincerely,

Ana Rita Gigante

19-Jan-2023

Reviewer 2 Report

Thank you for your revisions. There are still several instances of the term "mutations" in the text. These can easily be identified by searching the document (control + F) for "mutation". I leave it up to the editors regarding use of this terminology. Of course, any occurrence in the title of a reference should not be altered.

I would consider a little more robust discussion of the limitations. I agree that there is some novelty in what is likely the first report of pattern of familial ILD from Portugal. It is also understandable that resources have limited the evaluation and data collection on these patients somewhat. A more thorough, focused discussion of these helps put the results in proper context without diminishing them. Again, I will defer to the editors about whether this should be included in the final published manuscript, and to what extent. 

Author Response

We would like to thank you for your comments.

Thank you for your revisions. There are still several instances of the term "mutations" in the text. These can easily be identified by searching the document (control + F) for "mutation". I leave it up to the editors regarding use of this terminology. Of course, any occurrence in the title of a reference should not be altered.

We have changed all terms in the text as we see fit.

I would consider a little more robust discussion of the limitations. I agree that there is some novelty in what is likely the first report of pattern of familial ILD from Portugal. It is also understandable that resources have limited the evaluation and data collection on these patients somewhat. A more thorough, focused discussion of these helps put the results in proper context without diminishing them. Again, I will defer to the editors about whether this should be included in the final published manuscript, and to what extent.

We would like to thank the reviewer again for taking the time to review our manuscript.

Sincerely,

Ana Rita Gigante

19-Jan-2023